# Conceptual Model Modification and the Millennium Drought of Southeastern Australia

Justin Hughes [1,*], Nick Potter [1], Lu Zhang [1] and Robert Bridgart [2]

1   CSIRO Land and Water, Canberra, ACT 2601, Australia; nick.potter@csiro.au (N.P.); lu.zhang@csiro.au (L.Z.)
2   CSIRO Land and Water, Clayton South, VIC 3169, Australia; robert.bridgart@csiro.au
*   Correspondence: justin.hughes@csiro.au; Tel.: +61-2-6246 5962

**Abstract:** Long-term droughts observed in southern Australia have changed relationships between annual rainfall and runoff and tested some of the assumptions implicit in rainfall–runoff models used in these areas. Predictive confidence across these periods is when low using the more commonly used rainfall–runoff models. Here we modified the GR4J model to better represent surface water–groundwater connection and its role in runoff generation. The modified model (GR7J) was tested in 137 catchments in south-east Australia. Models were calibrated during "wetter" periods and simulation across drought periods was assessed against observations. GR7J performed better than GR4J in evaluation during drought periods where bias was significantly lower and showed improved fit across the flow duration curve especially at low flows. The largest improvements in predictive performance were for catchments where there were larger changes in the annual rainfall–runoff relationship. The predictive performance of the GR7J model was more sensitive to objective function used than GR4J. The use of an objective function that combined daily and annual error produced a better goodness of fit when measured against 80, 50 and 20 percent excedance flow quantiles and reduced evaluation bias, especially for the GR7J model.

**Keywords:** drought; rainfall–runoff model; non-stationarity; groundwater–surface water connection; objective functions

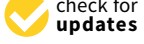

## 1. Introduction

Conceptual rainfall–runoff models are, at their simplest, a means to transfer a precipitation time series into a runoff hydrograph. They require calibration with observed time series and may or may not attempt to explicitly represent hydrological processes thought to be of interest. However, such is the need to estimate runoff from given rainfall, that their use is widespread and their predictive capacities are employed in environments or situations unforeseen at the time of model formulation. As new hydrological observations and knowledge becomes available, assumptions implicit in rainfall–runoff models can be tested and structural adjustments made.

The term "non–stationarity" is often used in the hydrological community [1], but can have various interpretations. In the strictest sense stationarity is the quality of time series data in which its probabilistic behavior does not change with time. In recent times long term droughts have seen changes in the annual rainfall–annual runoff relationship (alternatively expressed as runoff coefficient) in Australia [2–4] and China [5]. The phrase, "non-stationarity", tends to be used in relation to these observations since a single rainfall–runoff relationship becomes an inadequate predictor of annual runoff following extended drought. It should be stated that these observations do not necessarily imply a change in runoff generation processes, but at least they highlight our lack of understanding of such processes and how they relate to predictive models. Prolonged drought has allowed new insights into hydrological processes, even in areas where processes were thought to be well understood, e.g., Grigg and Kinal [6]. This is especially important for conceptual rainfall–runoff models since many processes affecting runoff generation are unknown

or ignored, but may be captured implicitly by the calibrated parameters. Any change in conditions between calibration and prediction can cause problems since the effects of such changes are not explicitly accounted for. So, in the context of rainfall–runoff models, non-stationarity is partly a problem of model structure and what processes it can represent (or not represent). Changes in land use, temperature, or atmospheric $CO_2$ levels may change the way in which vegetation uses water [7]. Changes in groundwater–surface water connection will affect the nature of runoff generation with implications for runoff estimation [2,8–10]. A multi–year drought in south–eastern Australia has significantly changed the rainfall–runoff relationship across around 37% studied catchments, leading to reduced runoff from a given amount of rainfall [3,11]. Internationally, many similar observations have been made in China [5,12], India [13] and the United States [14]. Conversely, a decrease in rainfall has resulted in observed streamflow increases in Sahelian Africa which remain poorly understood [15].

　　The utility of rainfall-runoff models across periods of climatic change (particularly extended drought) has been shown to be poor [16]. Analysis of model parameters calibrated prior to and within drought periods in SE-Australia indicates that some parameters that controlled infiltration and soil moisture storage were significantly different for the two periods [17]. Similarly, in SW-Australia, Silberstein et al. [18] suggest that despite the use of combinations of five different rainfall–runoff models, there was a pervasive error drift in all models towards the latter period (increasingly dry), indicating a change in state not captured by the models. In a separate study, Westra et al. [19] found that model fit prior to and during the "Millennium drought" (1997–2009) was improved by adding terms to the model that allowed the size of the watershed soil water accounting store to vary with an annual signal and a time trend across the simulation, i.e., a time varying parameter. Furthermore, conceptual rainfall–runoff models form the basis of the river system and planning models used by jurisdictions across Australia for allocation of resources and policy assessment. When these are applied outside of calibration conditions, as happened during the Millennium drought of south-east Australia, predictions were often poor with consequences for water users.

　　The study of Hughes et al. [20] attempted to address some the perceived shortcomings in model structure of the GR4J model [21] for the context of south–western Australia. The authors modified the production store in such a way that allowed the evaporative process to have a different sensitivity to storage than runoff production. This included a storage threshold below which evaporation could continue while runoff ceased. The modification helped to reduce long-term model error trends by producing a more pronounced declining trend in model storage and runoff in a drying climate. These modifications were examined in detail in a single research catchment with long-term ground water and streamflow observations [4], where model storage showed a strong correlation with catchment saturated volume. These structural changes were inspired by long-term observations of streamflow and groundwater storage across a period of extended climatic drying in south-western Australia where annual runoff showed a strong correlation with catchment groundwater levels in a highly non-linear relationship. The non-linearity in the groundwater–runoff coefficient relationship shows a strong influence of proximity of the water table in the riparian zone. When the water table is within 0–2 m of the riparian ground surface for at least part of the year, runoff coefficients (Q/P) are higher and very sensitive to the groundwater elevation. When the water table is disconnected from the riparian surface, runoff coefficients are much lower and have less sensitivity to groundwater elevation. Reductions in rainfall in that area across the last 40 years, and in particular the last 20, have seen large declines in runoff coefficient [2,8,9]. Few observations have been made where catchment storage increases due to the recent climate. However, increasing catchment storage (via forest clearing) has resulted in large increases in runoff coefficient following "re-connection" of the groundwater with the ground surface, sometimes many years following forest clearing [22]. Such observations provided conceptual understanding for the formulation of new hydrological models such as "LUCICAT" [23]. In particular,

the importance of groundwater–surface water connection was explicitly acknowledged in the LUCICAT model structure. Similarly, Deb. et al. [24], utilised a groundwater model coupled to a distributed surface water model to better represent "disconnection" and and reduced runoff during drought periods. Despite such advances, similar structural changes have not been evident in rainfall–runoff models.

The aim of this study is to test the benefit of those structural changes suggested by Hughes et al. [20] and Grigg and Hughes [4] on catchments in SE-Australia that may show changes in the rainfall–runoff relationship throughout drought periods. In particular, the ability of a new model structure to perform better via sustained storage depletion across extended drought (as suggested by Hughes and Vaze [25]). More specifically, this study will calibrate GR4J and a modified version of this with seven parameters (GR7J) during "wetter" periods and forecast into one or more "drier" periods to see if the modifications improve model predictability. This is a departure from the more usual split sample testing regime [26], since the experiment is specifically designed to test model performance outside of the calibration conditions. The pretext here is that more severe and extended droughts may be experienced in the future with no opportunity to calibrate models in transition to those conditions and, potentially, modifications to rainfall–runoff models may better equip them for such a transition.

## 2. Site Information

The study area and catchments used for this study are the same as those used by Saft et al. [3]. The catchments selected are largely free of contriving factors such as large reservoirs, substantial irrigation diversion or land use change. The catchments are located in close proximity to the great dividing range along the south-eastern coastline of the Australian mainland. These areas produce much of the runoff in south–eastern Australia including the Murray–Darling basin (MDB, Figure 1). Median annual runoff for the study catchments is 155 mm, while median annual precipitation is 870 mm and median annual potential evaporation is 1165 mm (Figure 2) In general, potential evaporation exceeds rainfall. Actual evapo-transpiration will vary more, since land-use across the study areas varies from forest in the more humid areas (generally greater than 800 mm p.a.), to pasture and annual crops, which dominate in the drier catchments. Seasonality of rainfall varies from north to south. In the south, rainfall is winter dominant, while in the north, summer rainfall is dominant.

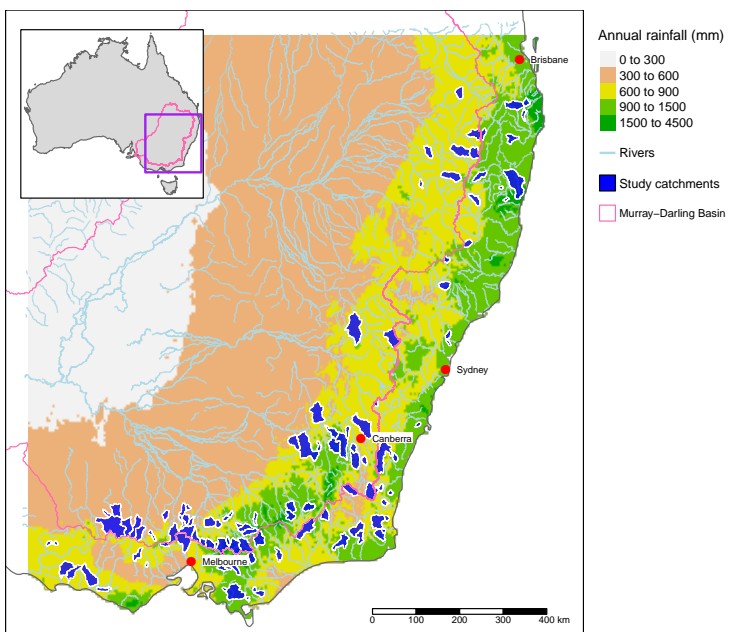

**Figure 1.** The study area showing study catchments and spatial distribution of mean annual rainfall.

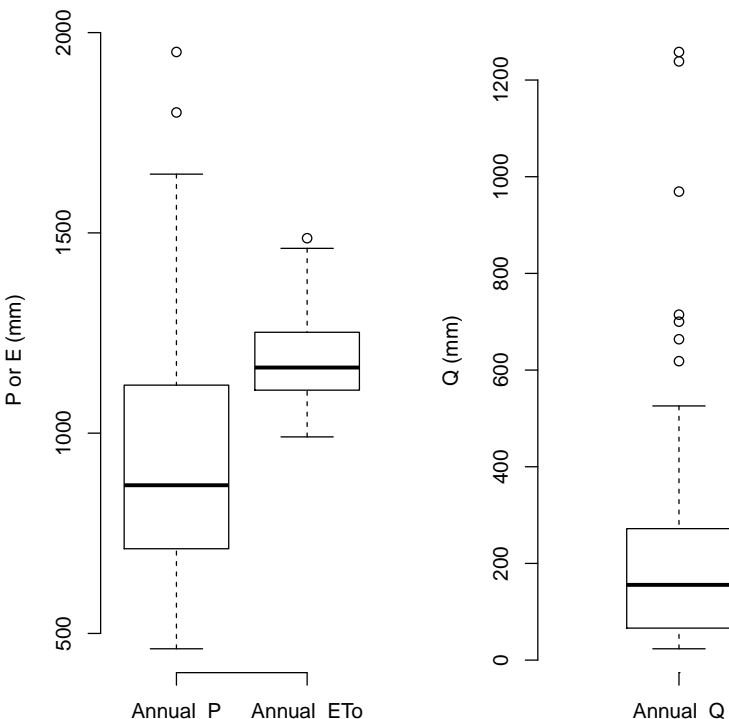

**Figure 2.** Climate and runoff boxplots for the study catchments. Boxes indicate 0.25 to 0.75 quantiles in data and whiskers indicate 1.5 times interquartile range.

## 3. Method

### 3.1. Drought Classification

This study uses the catchments and drought classification of Saft et al. [3]. Drought years were classified using annual rainfall. Annual rainfall anomaly data were calculated relative to the annual mean. The anomaly series was divided by the mean annual rainfall and smoothed with a 3-year moving window. Initially, all periods of consecutive smoothed negative anomalies were identified. To reduce the blurring effect of the moving window the exact end date of the dry period was determined through analysis of the un-smoothed anomaly data from the last negative 3-year anomaly. The end year was set as the last year of this 3-year period unless:

1.  there was a year with a positive anomaly >15% of the mean, in which case the end year is set to the year prior to that year;
2.  if the last two years have slightly positive anomalies (but each <15% of the mean), in which case the end year is set to the first year of positive anomaly

The start year of the drought period had not been adjusted in a similar way to allow for abrupt or gradual onset of the drought. The first year of the drought remained the start of the first three-year negative anomaly period. To ensure that the dry periods are sufficiently long and severe, in the subsequent analysis we only used dry periods that were 7 or more years long, and, with a mean dry period anomaly of $<-5\%$.

### 3.2. Magnitude of Change in Rainfall–Runoff Relationship Due to Drought

The magnitude of the change in the rainfall–runoff relationship due to drought was calculated by Saft et al. [3] using the method of Potter et al. [11]. Briefly, annual runoff data were normalised using a Box–Cox transformation [27], since the annual rainfall–runoff relationships are often non-linear, especially in drier climates. The transformed data were regressed against annual rainfall;

$$\hat{Q}_y = \alpha I_{wet} + \gamma I_{dry} + \beta P_y + \varepsilon_y \tag{1}$$

where $\hat{Q}_y$ is the transformed streamflow for year $y$, $\alpha$, $\beta$ and $\gamma$ are parameters of the linear model. $I_{wet}$ and $I_{dry}$, are non-drought (wet) and drought indicator variables (each assigned a value of 0 or 1 depending on the classification of the year). The magnitude of the difference between the drought and non-drought period was calculated using a characteristic drought rainfall for each catchment. This value was mean of the minimum and mean annual rainfall for the catchment. The expected values of runoff for a non-drought period and a dry period were calculated using fitted regression, and back transformed then differenced. This value was divided by the runoff for characteristic rainfall using the wet period regression to give a relative shift in runoff due to drought.

### 3.3. Model Framework

The GR4J model was used as a test-bed for model structural changes. The GR4J model [21], is widely used and relatively simple in nature. It consists of two main stores (Figure 3): the production store ($S$), and the routing store ($R$). Net rainfall ($P_n$) and net evaporation ($E_n$) were calculated as follows;

$$P_n = max(0, P_i - E_i) \tag{2}$$

and

$$E_n = max(0, E_i - P_i) \tag{3}$$

where $P_i$ was the precipitation on day $i$ and $E_i$ was the potential evaporation on day $i$.

$P_s$ is the amount on net rainfall that is added to the production store. This amount depends upon both net rainfall $P_n$, and the current state of the production store $S$;

$$P_s = \frac{x_1(1 - (\frac{S}{x_1})^2)tanh(\frac{P_n}{x_1})}{1 + \frac{S}{x_1}tanh(\frac{P_n}{x_1})} \tag{4}$$

where $S$ is the current level of storage (mm), $P_n$ is the net precipitation (mm), and $x_1$ is the maximum storage of $S$, and is a calibrated parameter. Net rainfall in excess of $P_s$ is routed to runoff;

$$P_r = P_n - P_s \tag{5}$$

where $P_r$ is the amount of net rainfall that becomes available for routing to runoff.

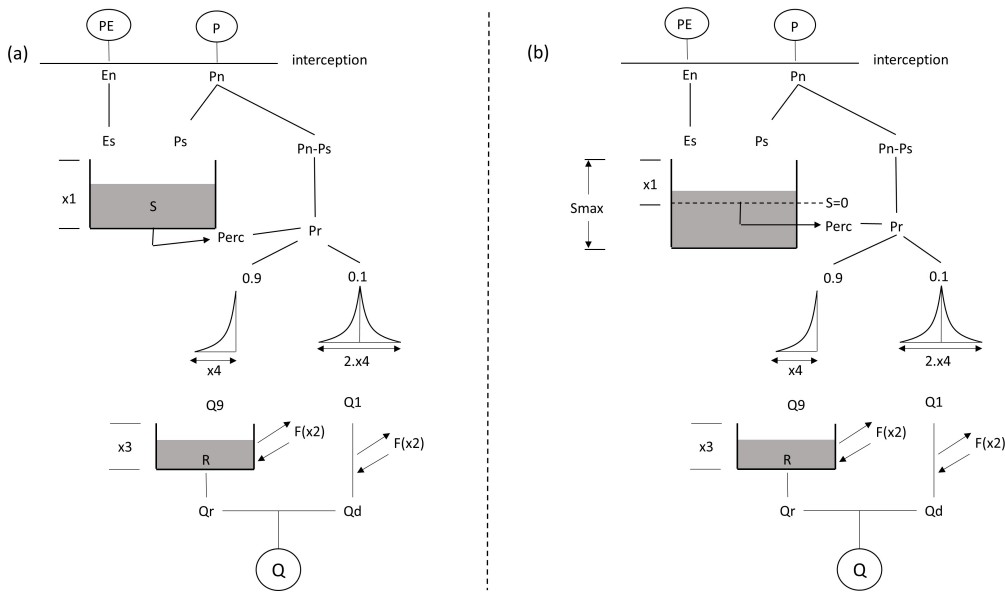

**Figure 3.** Model structure conceptual diagrams for (**a**) GR4J and (**b**) GR7J.

In a similar way, evaporative loss from the production store ($E_s$) is calculated relative to the level of the production store ($S$);

$$E_S = \frac{S(2 - \frac{S}{x_1})tanh(\frac{E_n}{x_1})}{1 + \frac{S}{x_1}tanh(\frac{E_n}{x_1})} \tag{6}$$

where $E_n$ is net evaporation (Equation (3)). The GR4J model code was altered with the aim of allowing more flexibility in the way the processes encapsulated in Equations relate to the state of the production store ($S$).

To vary the sensitivity of $P_s$ to $S$, the exponent in the numerator of Equation (4) was re-coded as a calibrated parameter of value between 1 and 3;

$$P_s = \frac{x_1(1 - (\frac{S}{x_1})^{x_5})tanh(\frac{P_n}{x_1})}{1 + \frac{S}{x_1}tanh(\frac{P_n}{x_1})} \tag{7}$$

where $x_5$ is the new calibrated parameter. Such a change can, for example, allow $P_s$ to reduce as $S$ approaches the value of $x_1$, i.e., as the production store becomes full more less net rainfall contributes to the production store and more goes into the routing store to produce runoff.

The updated $E_s$ calculation required two additional calibrated parameters ($x_6$–magnitude factor and $x_7$–shape factor), and a user defined value (*expand*) which determines the size of the available depletion relative to the size of the production store available for runoff production ($x_1$);

$$E_s = E_n * x_6 * \left(\frac{(S_{max} - x_1 + S)}{S_{max}}\right)^{x_7} \tag{8}$$

where $S_{max} = expand * x_1$ and *expand* have a value greater than or equal to 1. Effectively, a value of *expand* $> 1$ allows depletion of the production store to continue via evaporative processes when all incoming precipitation $P_n$ is captured by the production store and none is routed to streamflow. In this study an *expand* value of 2.0 was used. Initial investigation suggested that such a value would have some predictive benefit in regards to ephemeral streams in particular.

It should be noted that no modifications have been made to the routing store calculations, and these remain identical for GR4J and GR7J. The modifications detailed above are intended to better encapsulate longer term process, whereas routing functions are more focussed on shorter time-scales which are beyond the scope of this study.

### 3.4. Model Calibration

Both GR4J and GR7J were calibrated in identified wet years with evaluation occurring in identified drought years. Goodness of fit was compared for GR4J and GR7J in the evaluation period. All models were calibrated with the Shuffled Complex Efficiency algorithm [28]. Models were calibrated with two different objective functions. The first was a linear combination of Nash–Sutcliffe Efficiency (NSE [29]), using root-transformed values and total simulation bias:

$$NSE_B = \left(1 + \frac{\sum_{i=1}^{n}(Q_{obs,i}^{\lambda} - Q_{sim,i}^{\lambda})^2}{\sum_{i=1}^{n}(Q_{obs,i}^{\lambda} - \overline{Q_{obs}^{\lambda}})^2}\right) * \left(1 + \frac{|\sum_{i=1}^{n}(sim) - \sum_{i=1}^{n}(Q_{sim,i})|}{\sum_{i=1}^{n}(Q_{obs,i})}\right) \tag{9}$$

where n is the number of time steps, $Q_{obs,i}$ and $Q_{sim,i}$ are the observed and simulated flows, and $\lambda$ is the power value for the transformation of data.

Systematic model errors can be observed when the sign of the error persists over a series of time intervals [30]. When using total bias as a, or part of, an objective function, long runs of consecutive equal sign error are possible even when total bias is low. This is possible since, in most cases, the error residual is auto-correlated, indicating that the model has not adequately described all processes evident in the data. To counter this

tendency, a second objective function was used where absolute bias was calculated for 365 day segments of the simulated and observed time series. The mean of the absolute bias of these segments was combined with NSE in the following objective function:

$$NSE_{Seg} = \left(1 + \frac{\sum_{i=1}^{n}(Q_{obs,i}^{\lambda} - Q_{sim,i}^{\lambda})^2}{\sum_{i=1}^{n}(Q_{obs,i}^{\lambda} - \overline{Q_{obs}^{\lambda}})^2}\right) * \left(1 + \sum_{k=1}^{m}\left|\sum_{j=1}^{365}\frac{(Q_{sim,j} - Q_{obs,j})}{\sum_{j=1}^{365}Q_{obs,j}}\right| * \frac{1}{m}\right) \quad (10)$$

where $m$ is the number of 365-day segments the simulated and observed time series are subdivided into. For this study the $\lambda$ value used was 0.5. The scale of aggregation was chosen to be 365 to approximate annual bias; however, other time scales could be investigated. Similarly, the weighting of terms in the objective functions is assumed to be even. Further investigation of differential weighting schemes may be beneficial.

Evaluation goodness of fit was expressed in terms of NSE and total bias. Evaluation goodness of fit for combinations of model and objective function were compared to GR4J using the Wilcoxon test for significance [31].

## 4. Results

Two main drought periods were identified using the drought identification method (Section 3.1) across all rainfall data. The extended drought was identified in the majority of catchments during the mid 1990's to 2009. This period is commonly referred to as the "Millennium drought". In other catchments, where the time series is long enough, the "World War II drought" around 1940 is also classified as drought. These can be seen in Figure 4. The drought magnitude is calculated using the characteristic drought rainfall. This is substituted into Equation (1), then the difference in runoff values from drought and non-drought years is calculated. The mean drought runoff reduction due to non-stationarity in the rainfall–runoff relationship was 30%, although ~25 catchments showed an increase in drought periods. For more information see Saft et al. [3].

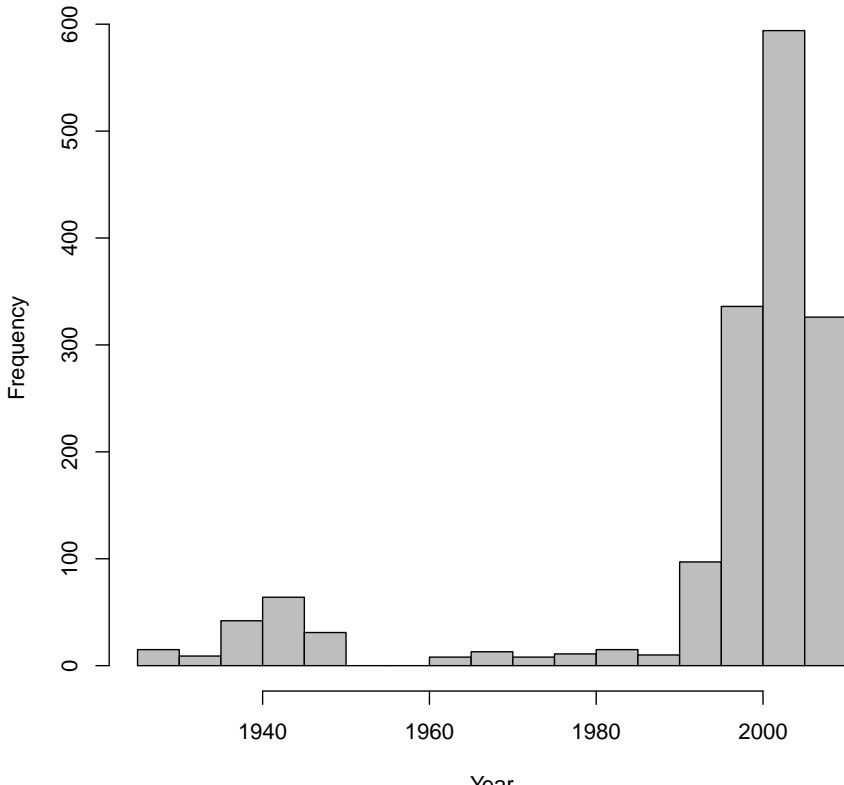

**Figure 4.** Histogram of identified drought periods across all catchments.



Calibration of the two models resulted in very similar values for both NSE and bias (Table 1). Slightly better calibration bias was apparent for the $NSE_B$ objective function, although this is not surprising since the $NSE_B$ objective function is, in part, a total bias calculation. Despite calibration scores of very similar values, the GR7J model, when used with appropriate objective functions, shows an improved fit across the flow duration curve relative to GR4J. Figure 5 shows relative error for models at 20, 50 and 80 percentile exceedance flows calculated across the calibration period. GR7J was clearly superior at low flows (80% exceedance). The results indicate that the GR7J model is more sensitive to the objective function used and can achieve lower error than other model and objective function combinations. NSE scores of all models in the evaluation period were very similar, with a slight degradation in NSE for the GR7J/$NSE_{Seg}$ combination. Improvements in evaluation bias were apparent for GR7J over GR4J. This was especially so for the GR7J/$NSE_{Seg}$ combination that had a mean bias improvement of 14.9% over the next best model/objective function combination. The significance of the improvement in evaluation bias was tested using a Wilcoxon test for model combinations relative to GR4J/$NSE_B$ (Table 2). Assuming that a *p* value of less than 0.05 indicates a significant difference, the evaluation or drought period bias for GR7J is significantly lower than GR4J/$NSE_B$ for the case of GR7J/$NSE_{Seg}$ where bias is 44% lower than GR4J/$NSE_B$ at *p* = 0.004. These results also suggest that an appropriately formulated objective function is more important as the number of model parameters is increased.

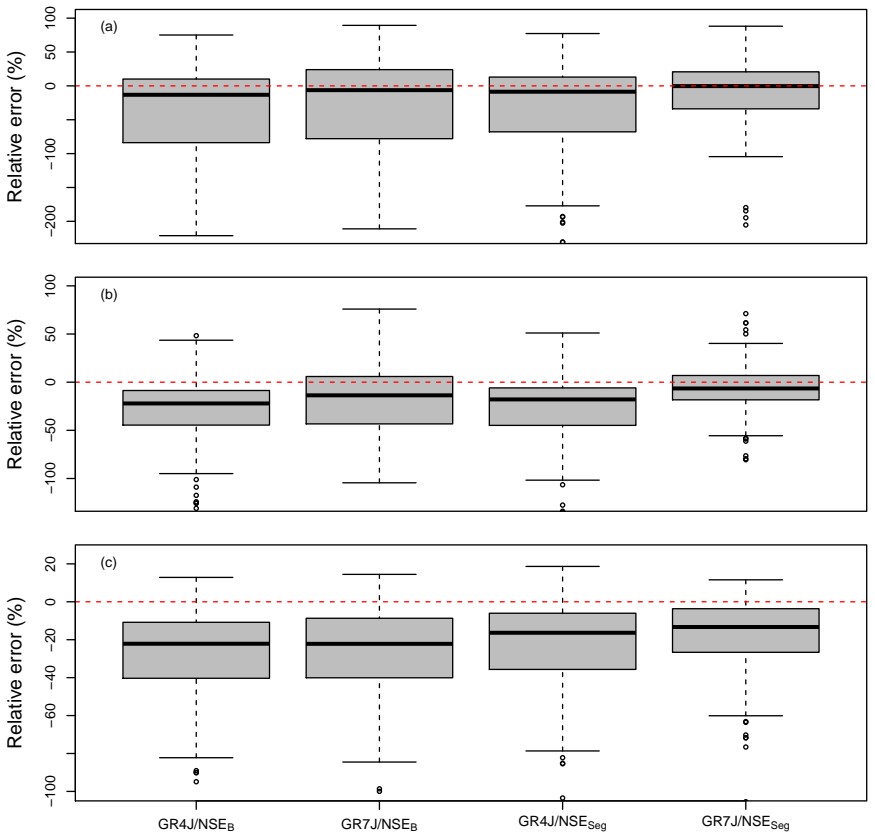

**Figure 5.** Prediction error of combinations of model and objective function at (**a**) 80% exceedance, (**b**) 50% exceedance and (**c**) 20% exceedance quantiles. Error is expressed as a percentage difference from the corresponding observed value. Boxes indicate 0.25 to 0.75 quantiles in data.

**Table 1.** Mean model goodness of fit during calibration and evaluation (drought) periods for the four model-objective function combinations.

| Period | Goodness of Fit | GR4J | | GR7J | |
|---|---|---|---|---|---|
| | | $NSE_B$ | $NSE_{Seg}$ | $NSE_B$ | $NSE_{Seg}$ |
| Calibration | NSE | 0.71 | 0.70 | 0.76 | 0.72 |
| | Absolute bias(%) | 0.1 | 5.6 | 0.0 | 4.4 |
| Evaluation | NSE | 0.64 | 0.62 | 0.66 | 0.59 |
| | Absolute bias(%) | 59.3 | 47.9 | 50.0 | 33.0 |

**Table 2.** Wilcoxon test $p$ values for evaluation period goodness of fit relative to GR4J/Nash–Sutcliffe Efficiency $(NSE)_B$.

| | GR4J | | GR7J | |
|---|---|---|---|---|
| | $NSE_B$ | $NSE_{Seg}$ | $NSE_B$ | $NSE_{Seg}$ |
| NSE | NA | 0.507 | 0.596 | 0.340 |
| Absolute bias(%) | NA | 0.110 | 0.883 | 0.004 |

The magnitude of change in the rainfall–runoff relationship during extended drought was calculated for each catchment and expressed as a percentage change relative to the non-drought period. This was then compared to the improvement in evaluation (drought) bias for the GR7J/$NSE_{Seg}$ model over the GR4J/$NSE_B$ to understand if the GR7J model structure is more effective in coping with non-stationarity in the rainfall–runoff relationship. This is illustrated in Figure 6, showing increased benefit of GR7J in terms of evaluation bias as the relative change in the rainfall–runoff becomes increasingly negative (drier). The superior fit of GR7J at low flows (Figure 5a), will result in superior fits in drought conditions as these will obviously be at higher exceedance flow values. Where the magnitude of change in the rainfall–runoff relationship is positive or close to zero, there appears to be little predictive benefit in the GR7J formulation. This is also evident spatially, where the benefit of the GR7J/$NSE_{Seg}$ models are generally located in those catchments described by Saft et al. [3] as catchments with "change detected", in reference to the effect of drought on rainfall–runoff relationships (Figure 7-visible as catchment polygons with blue fill).

Increasing dryness and disconnection of groundwater and surface water can be indicated by the increasing frequency of no-flow days [32]. The predictive advantage to GR7J showed some influence in the number no flow days experienced within the evaluation period (Figure 8), and the authors consider this is a reasonable validation of the structural changes tested.

GR7J has the facility to allow the evaporative and runoff generation processes to have different sensitivities to store levels, and enable evapo-transpiration to continue when production store levels are below those at which no incoming net rainfall is moved to the routing store. This is visible as a negative value in the production store plots (Figure 9). When droughts are encountered, production store levels become negative. In this situation, the routing store will only receive new contributions of water when the value of net precipitation (input) is of higher magnitude than the production store deficit. Such a mechanism can reduce flows to near zero levels and will not produce a runoff response to small volumes of net precipitation. This also has the effect of creating a deficit and reducing runoff volumes as catchment storage transitions from dry to wet. This is illustrated in Figure 9 that plots production store levels and runoff for a single catchment in the year 1983. The year 1982 was a severe drought, and production storage was low in early 1983. GR4J overestimates runoff for most of the year, while GR7J slightly underestimates runoff; however, it can be seen that a deficit (production store) resulting from drought carried forward to the following year is effective in reducing early season flows and achieving a better match to observations.

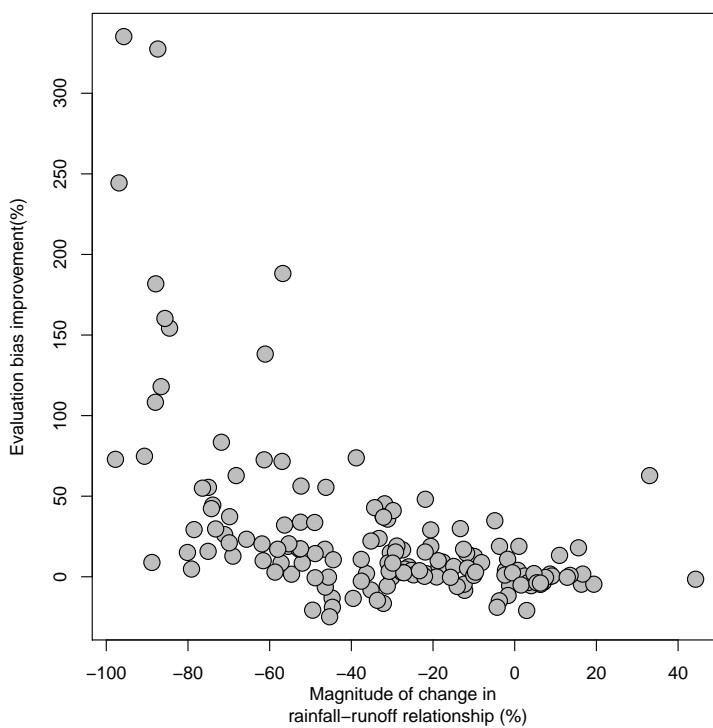

**Figure 6.** Evaluation period bias improvement of GR7J over GR4J in relation to the magnitude of the change in the rainfall–runoff relationship in drought periods.

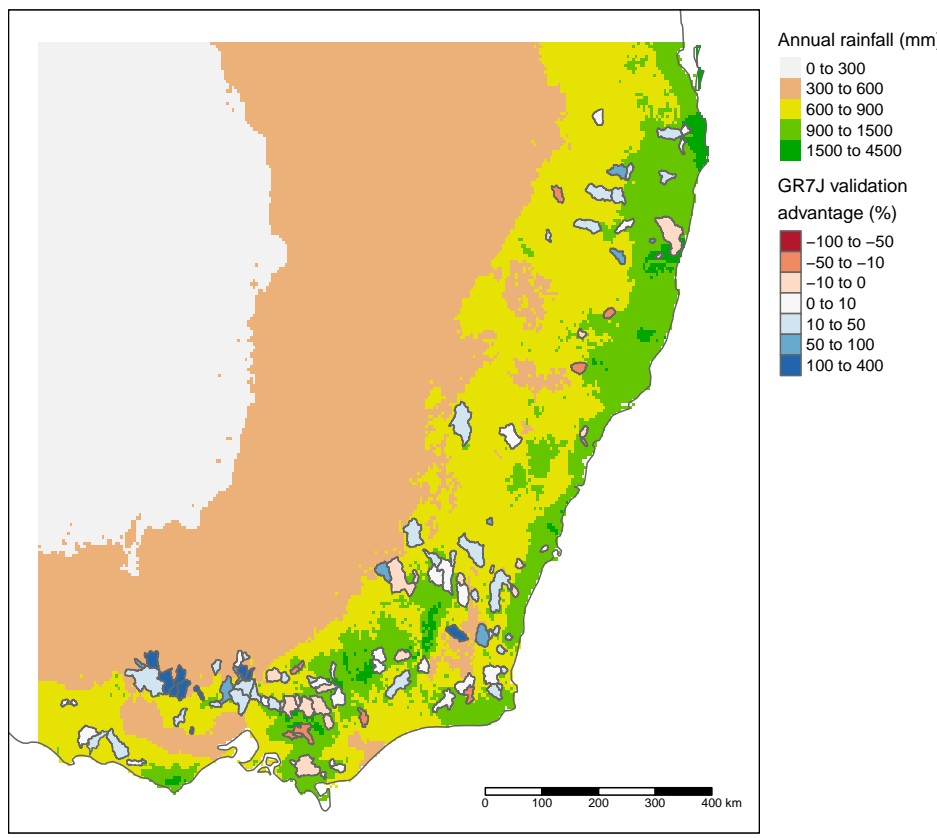

**Figure 7.** Catchment location and evaluation period bias improvement of GR7J over GR4J.

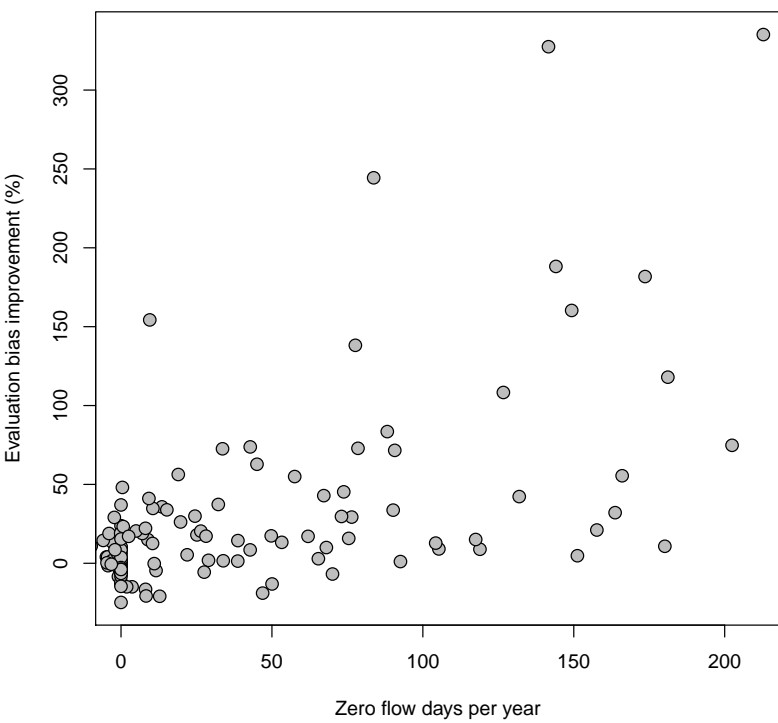

**Figure 8.** Evaluation period bias improvement of GR7J over GR4J in relation to the increase in mean annual zero-flows days from the calibration period to the evaluation period.

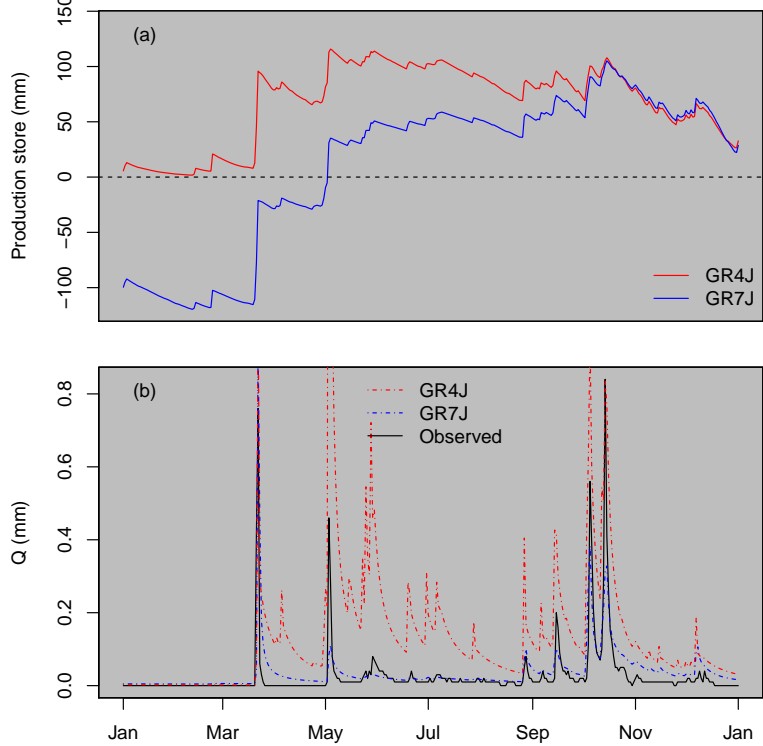

**Figure 9.** Production store levels (a) and runoff(b) for calibrated GR4J and GR7J models of catchment 222,007 in the year 1983.

## 5. Discussion

The GR7J model shows an improved predictive performance in long-term drought periods relative to GR4J, at least in terms of total bias, and error at selected quantiles. In many catchments studied here, annual runoff cannot be predicted well during drought periods using a single annual rainfall–runoff relationship [3], indicating some process change, or the expression of processes at an extreme. Similar observations have been made in south-western Australia [2] through a period of continued drying over 35 years. Calibrating rainfall–runoff models during relatively wet conditions and predicting during dry conditions has shown to be difficult [16]. The model structure of GR7J was intended to better cope with extended periods of drought. The inclusion of a threshold in the model production store was intended to allow the evaporative process to continue after streamflow was minimal, producing model deficits that needed to be recovered before runoff could potentially recommence. Such a structure has shown to be beneficial here as it has in the study of Grigg and Hughes [4]. This will obviously have benefits in ephemeral catchments where long periods of no flow are common and become extended in drought. The predictive advantage for GR7J was greater where the rainfall–runoff relationship shifted strongly during drought. (Figure 6). However, it was noted that while the GR7J code could reduce error in low flow and drought periods, over-prediction was still apparent. The changes implemented provided genuine improvements and could provide the basis for further model improvement [25].

It should be noted here that the method of quantifying the shift in rainfall–runoff relationship used in this study considered a change in the intercept term of the relationship without consideration given to any change in slope (or any interaction between intercept and slope). It is, therefore, possible that inclusion of such terms in the detection method would change the magnitude or relationship changes used in this study. The slope term was tested in the study of Saft et al. [3], but was not used in the final manuscript for simplicity and since relative magnitudes of change were not greatly different (N. Potter pers. comm.). However, it is obvious in some studies e.g., Grigg and Hughes [4], that slope of the rainfall–runoff relationship undergoes marked changes and, for these locations, the method of Saft et al. [3] may be inadequate.

The changes implemented in GR7J, were inspired by long-term observations of climate, rainfall, and groundwater information in the south-west of Australia. There exists some similarity in runoff generation processes between south-western Australia, and the southeast, at least with regard to the amplifying effect of the water table being close or at the ground surface. Hydrometric and tracer studies across the MDB indicate that groundwater proximity to the surface not only provides runoff directly into streams, but is a critical factor in promoting saturated overland flow [33–36]. This behaviour also implies a distinct threshold in the relationship between catchment storage (more specifically the water table) and runoff generation. As the water table falls well beneath the ground surface in extended drought, there will be less possibility of direct groundwater flow into stream channels and saturated overland flow. In these circumstances there will be increased capacity for the catchment to store incoming rainfall in the unsaturated zone at the cost of runoff. This water can be subsequently lost to evapo-transpiration. While GR7J has a structure that may offer some predictive improvement, these processes are not well represented and may be worth further experimentation.

A major limitation of conceptual rainfall–runoff models is the tendency to represent catchment systems as one-dimensional storage and routing schemes. As such, better representation of the three dimensional and multi-temporal nature of catchment hydrology may require more complicated hydrological models. This situation is highlighted by catchment drying in southern Australia. Models that can explicitly represent the influence of topography, land use and groundwater–surface water connection may be of higher value in the future, whereas in a more stable climate, simpler models may offer advantages.

Characterising the performance of models is a critical component in the field of hydrology. For a comprehensive review on the topic see Bennett et al. [37]. A genuine

issue in calibrating rainfall–runoff models is the form of the objective function, since these models need to be calibrated. The error residual is the general basis for many objective functions [29]. However, many problems associated with objective functions stem from the assumptions made about the underlying properties of the error. If the objective function is based upon variance (e.g., NSE), the errors must have a mean of zero, be homoscedastic, be normally distributed and pairwise uncorrelated. However, these conditions are rarely met [38]. Aitken [30] acknowledged that autocorrelation in errors required specialised objective functions to reduce over-fitting. With respect to the Millennium drought in SE Australia, the importance of objective function to model performance was also demonstrated by Fowler et al. [39], using a similar approach to Kim et al. [40], i.e., segmenting the model bias across time to reduce problems related to auto-correlated error.

The objective functions chosen for this study calculated error across two temporal scales. Both calculated daily error by the use of NSE. Additionally, $NSE_B$ calculated total error using total bias, while $NSE_{Seg}$ calculated error at an annual scale. While calculating total error would seem to help in an intuitive way, further examination showed that such a calculation could cause problems in optimisation. These problems occur when a portion of the simulation fits poorly to observed values. To reduce total bias, parameter sets are chosen that *compensate* for the period of poor fit, thereby providing inappropriate parameters for different sub-periods of the simulation time series. The use of the $NSE_{Seg}$ avoids this problem since mean absolute annual error is used which can isolate poorly performing periods more effectively, and reduce over-fitting at the daily scale. More sophisticated approaches have been suggested to cope with multiple scales of error detection. The use of wavelet analysis provides some potential in that regard [41,42], as does a sub-period calibration and evaluation [40]. Increasing the number of free parameters offers the opportunity to include more complexity in the model, but increases the opportunity for over-fitting. This was the case here since GR7J performed far better in evaluation when fitted with the $NSE_{Seg}$ objective function, which is considered a better balance of daily and annual error for calibration. While NSE alone is likely to over-fit at the daily scale, this is tempered by the annual absolute bias calculation in the $NSE_{Seg}$ objective function.

The value of increasing the number of free parameters in a model can be be judged against any improvements in performance by using measures such as Bayesian Information Criteria (BIC) or Akaike Information Criteria (AIC). Other approaches such as k-fold cross validation have been used. However, in this instance, we aim to test the model predictive performance in strict conditions (drought), making such approaches difficult. Furthermore, the approach used here demonstrated specific structural weaknesses in the GR4J model for this application and was inspired by hydrological observations. We used a "bottom-up" approach based on observations [4] rather than a "top-down" approach e.g., Pushpalatha et al. [43]. In a sense, the model modifications used here and previous studies [4] resulted from a close coupling of experimental data and an identified predictive deficiency. Such approaches have been shown to have value [44] but remain uncommon.

## 6. Conclusions

The structural modifications that produced the GR7J model have some predictive benefits in situations where there is a shift in the annual rainfall–runoff relationship to a drier state. One mechanism contributing to this is a threshold in the production store that allows evaporation to continue when storage available for streamflow production has been depleted. Moreover, the GR7J model allows the evaporative and runoff production processes to have a differing sensitivity to production store levels (apart from the threshold). The GR7J model had more sensitivity to the objective function used for calibration in terms of its predictive performance. A considerable predictive improvement was apparent for the use of an objective function incorporating daily and annual error, as opposed to daily and total error (for the GR7J model). GR4J predictive performance was less sensitive to objective



function. Future modification and increasing model complexity may require objective functions that operate at multiple temporal scales to deal with over-fitting problems.

The structural modifications tested here do assist in runoff production prediction during drought but, in general, still overestimate drought runoff. When considering observational studies across the MDB, it may offer some utility to rainfall–runoff models to include processes that better represent the groundwater–surface water connection and its strong influence on runoff generation.

**Author Contributions:** J.H. and N.P. propsed the methods and manuscript concept, R.B. wrote the model code and data routines, L.Z. revised the ideas and assisted with the literature. All authors have read and agreed to the published version of the manuscript.

**Funding:** This research received no external funding.

**Conflicts of Interest:** The authors declare no conflict of interest.

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
