# Peer review of "Conceptual Model Modification and the Millennium Drought of Southeastern Australia"

_water, doi:10.3390/w13050669_

Round 1

Reviewer 1 Report

In this well-prepared manuscript, the authors have evaluated the capacity of a previously developed model improvement for the GR4J model to predict streamflow in drought conditions. The authors add parameters to the GR4J to account for the disconnection of groundwater and streamflow in drought conditions with the hypothesis that this will better represent low-flow conditions. They provide evidence that documents this improvement. This manuscript is an incremental improvement over work by these same authors in that no new methods are presented, and this manuscript is worthy of eventual publication in that it demonstrates the advantages of the model improvement previously hypothesized on one basin by extending application to additional basins in drought condition. Prior to publication, this work requires a “major” revision.

The authors consistently use language suggesting that the drought has changed rainfall-runoff relationships (examples on lines 1, 39, 126, etc.). This is a question of the precision of language. It is incorrect to state that the drought caused a change. Rather, the drought conditions have highlighted a deficiency of the model conceptualization. The current language implies that the millennium drought caused a new pathway, implying that droughts prior to the millennium drought would have been adequately modeled under the previous GR4J assumptions. There is no evidence to support this claim. In this light, and as pedantic as it may seem, I think it more correct to say that the severity of the millennium drought demonstrated a weakness of the models hydrologic conceptualization (though perhaps you can say it more elegantly – ha ha).

The concern that necessitates a major revision revolves around Equation (1). This equation is, ultimately, used to represent the change in the rainfall-runoff relationship between wet and dry conditions (e.g., in Figure 6). For simplicity here, let Q_dry be Equation (1) using the dry parameters, such that Q_dry = gamma +beta*P. Similarly, let Q_wet = alpha + beta*P. In both cases, the P is characteristic drought condition. Leaving aside the question of back transform for a moment, the magnitude of change is given as (Q_dry – Q_wet) / Q_wet. Careful inspection shows that this simplifies to (gamma – alpha) / Q_wet. Because gamma and alpha are from a linear regression, they represent the standardized streamflow in the absence of precipitation (y-intercept). Therefore, the magnitude of change computed here – and shown in Figure 6 – is merely a difference in roughly average dry and wet conditions in the absence of precipitation.

If the rainfall-runoff relationship is truly changing, then we would expect that elasticity of precipitation to be changing rather than just the average no-P condition. This can easily be evaluated by modifying Equation (1) to correctly use indicator variables and to add an interaction term. The revision should be: Qhat_i = alpha + gamma1 * I_dry + beta * P_i + beta1 * P_i * I_dry + epsilon_i. The first point: Because I_wet and I_dry are mutually exclusive, you don’t need both. In wet conditions the constant is alpha and the slope is beta (same as yours); in dry conditions the constant is (alpha + gamma1) and the slope is (beta + beta1). The interaction term (P*I) allows the slope (elasticity) of precipitation to change. (Note, I use things like gamma1 to distinguish from your notation; chose a notation that works for you.)

Given this new formulation of Equation (1), the magnitude of change can be given as (gamma1 + beta1 * P) / Q_wet. This would demonstrate a complete change in the basin rainfall-runoff relationship, demonstrating that the null condition has changed as well as the elasticity. (It might even be that the elasticity change (beta1) is the most important term for your application.) The change is now a function of the characteristic drought precipitation. I recognize that Equation (1) is based on previous work by the authors; I hope this does not prejudice them against change. To make a claim that the rainfall-runoff relationship is changing, I think it important to show that the elasticity has changed. I cannot recommend publication without settling this point through a well-reasoned response or a change.

Turning to a less substantial point, I’d like the authors to add a brief discussion of parsimony. What objective methods demonstrate the change in model performance justifies the near doubling (from 4 to 7) of model parameters? I don’t disagree with the utility of this approach. Rather, I think the authors might strengthen their case by explicitly acknowledging that increased model complexity if worthwhile and demonstrates performance beyond what you would expect from simply additional parameterization.

MINOR COMMENTS:

Line 18: Hydrograph refers to flow, please find a different term for “precipitation hydrograph”.

Line 18: Not all models require calibration.

Figure 5: Please define the whisker ends. Are you using 1.5*IQR or some quantile? (I prefer the latter, but that doesn’t matter as long as you specify what you did.)

Reviewer 2 Report

Paper gives insight into a very interesting, actual, but thankless topic about drought modeling. Theoretical knowledge about modeling is applied for real case study, which I welcomed, especially without analysis of the time series, which is sometimes very hard to perform. I am suggesting a minor revision. Here are my comments.

1) Figure 1: can authors explain what is about inner areas where analysis was not provided?

2) Why was the classification of Soft used? Please, explain it. Same question about subchapters 3.2., 3.3. and 3.4. Application of a particular method should be elaborated. 

3) Why didn't the authors calculate drought indexes, for example SPI, EDI, SIAP?

Reviewer 3 Report

The first thing that irritates is that, as it seems, an unfinished, raw manuscript was submitted:

  • Keywords are missing. Also missing are Author Contributions, Funding, Acknowledgments and Conflicts of Interest.
  • The line numbering is not continuous, e.g. chapter 3.3
  • To name but a few, errors appear in the text, which should be noticeable with conscientious error correction.
    • Line 233 " ...it can bee..";
    • Line 263 "... in the the..";
    • line 195 "…indicates a significantly difference…”
    • Figure 5: "… form…”
    • Reference 8: "2015, In Press.

Thematically, the work is interesting, even if many passages already confirm widely accepted knowledge or the authors often confirm their own opinions. 40 per cent of the references refer to work by the authors themselves or with the authors' collaboration.

Concrete advice:

  • Line 26: In the strictest sense non-stationarity is the quality of time series data in which its probabilistic behavior does not change with time.
  • Line 47 and Ref.11: Silberstein not Silberstien
  • Line 82: “…similar structural changes have not been evident in rainfall - runoff models.” is a bit bold statement
  • Line 97: (MDB, Figure 1) For readers not so familiar with Australia, MDB is not recognisable in Figure 1.
  • Figure 2: It is advisable to use the same abbreviations for the same parameters throughout the text and to explain them consistently in the text (E; ETo; Pi; ei…
  • Equations 1,2,3: Index i is used on the one hand for year i but the same for day i
  • Figure 3: Routing storage R is not explained in the text. It should at least be said why this is not being addressed (x2, x3, x4, Q9, Q1…).
  • It would be advisable to swap figures 8 and 9 after the latter is mentioned as the former in the text.
  • Line 233 “ underestimates” seems inconsistent with line 253 “over-prediction”
  • Also, in the discussion, 8 out of 15 references are from the authors or their environment. Some of the remaining ones are standard or pure overview citations. It is recommended to complement the discussion with some international work, of which there are many in the field.
  • The relationship of the wavelet analysis, first mentioned in line 304, to the rest of the text is difficult to discern.
  • Reference 34: “parametereparameter”

Round 2

Reviewer 1 Report

I reviewed a previous version of this manuscript. The manuscript remains strong, though the responses to previous comments have failed to address major shortcomings of the manuscript. While I am open to disagreement, I cannot recommend publication of this work in its current form.

On the point of whether the drought has changed the relationship between rainfall and runoff. I accept and appreciate the evidence provided by the authors. The experiments and analysis presented by the authors does not provide sufficient evidence to determine if the relationship has change or if the new conditions are a failure of the model’s conceptualization. I am not asking for any further experiment; instead, I think this comment can be addressed by discussion. I am sure the authors can provide an interesting discussion on the literature and their opinion as to whether the phenomenon they demonstrate motivates a change in our conceptual modeling (new models needed to cover all conditions) or an update to the parameters of the conceptualization (new parameter values, same conceptual design). The authors responded well to this comment, and I was disappointed that the manuscript did not reflect changes.

On the topic of equation (1), the response to the comment reflects a resistance to consider a flaw in the manuscript. It is the response to this comment that prevents me from supporting publication at this time.

The authors acknowledge that the method in the manuscript is deficient but commit themselves to it regardless of the inherent flaw. In my previous comment, I noted that Equation (1) is deficient because it does not allow for a slope change under wet and dry conditions; it supports only a change in the intercept term. The authors concur with this argument but commit to using the method “rightly or wrongly” simply because the previous method is widely cited despite its shortcoming. They argue that if they resolve this conflict, then they could not compare their result to previous methods. This response is problematic because it commits to using a method known to be inappropriate and incorrectly suggests that correction would prevent comparison.

There are several possible solutions here that would address this flaw; here is one. The authors could acknowledge the shortcomings of the previous method through discussion, allowing them to demonstrate a deep knowledge of detecting change. The authors could also apply the method with an interaction term – as they acknowledge to be more appropriate – alongside the previous method. In this way, the authors could continue to cite the previous work, discuss how the previous work requires modification for this application, and demonstrate the change as presented by the original Eq. 1 and the modified version. The result would be a more defensible manuscript to prove their point.

This is a very strong experiment. I am confident that the correction of the flaw around Eq. 1 will not dramatically change the results and will not substantially change the conclusions. However, the equation, and the method, is deficient. As much as I find the work interesting and expect that the conclusions are correct, I cannot recommend publication on methods that the author acknowledge to be inappropriate (especially when the modification could be simple). It puts me, the reviewer, in a tricky situation: wanting this work to achieve publication and not being willing to accept flawed methods.

NOTE TO AUTHORS: As I hope you can detect my anguish above, not wanting to disparage a strong paper and not wanting to support inappropriate methods, I’ll note here that I was torn between “Reconsider after major revision” and “Reject” with a leaning towards the latter. Instead, I have selected the former in the belief that you are prepared to make meaningful changes to the manuscript and that the require changes would not be overly burdensome in the pursuit of accuracy. While I’m not asking you accept my suggestions without question, I am looking forward to revision of the manuscript that demonstrates a consideration of the flaws you acknowledge.

Round 3

Reviewer 1 Report

Thank you for taking the time to revise the manuscript.

The authors have revised to include discussions of the relevant concerns. I think this is acceptable.

To Eq. (1), I still disagree with the approach. The authors also seem to not be fully comfortable with the approach (as they consider it might be inappropriate) but are not committed to seeking another solution. While I think this limits the impact of the manuscript, I respect their decision. A minor note: The citation of personal correspondence with a co-author of the paper feels a bit odd. Perhaps just use the “we” to justify?

This is very good work. I am glad to see it in the literature. I appreciate the patience the authors have shown.